# Mapping Role-Playing Games in Ibero-America: An Educational Review

**Mario Grande-de-Prado** [1,*] **, Roberto Baelo** [1] **, Sheila García-Martín** [1] **and Víctor Abella-García** [2]

1   Department of General and Specific Didactics and Educational Theory, University of León, León 24071, Spain; roberto.baelo@unileon.es (R.B.); sgarcm@unileon.es (S.G.-M.)
2   Department of Educational Sciences, University of Burgos, Burgos 09001, Spain; vabella@ubu.es
*   Correspondence: mgrap@unileon.es

**Abstract:** Role-playing games (RPGs) have a controversial public image in several countries, including Spain. These fears lack a scientific basis, as role-playing games may be useful in education. Educational trends such as gamification are helping to change this perspective, incorporating elements of RPGs in applications like learning management systems (LMS), e.g., Classcraft. Given the increased research interest in this topic, this paper presents a systematic literature review (SLR) report on the state-of-the-art related to RPGs in an Ibero-American education research context. In the study, a comprehensive search is carried out for the most relevant research papers indexed in Latindex, founded through the virtual repository Dialnet for papers between 2010 and 2019 in the field of education. The search chain was 'role-playing games', erasing those topics not related. Results show that there are several relevant references, even though they do not seem to have had a great impact. It can be concluded that there is an interest in RPGs in education, especially in Spain, but their potential is still to be developed.

**Keywords:** role-playing games; game-based learning; gamification; mapping

## 1. Introduction

Today's education is developing within a plural and complex society, in addition to being characterized by rapid mutability [1] in a volatile, uncertain, complex, and ambiguous (VUCA) world [2]. This implies that at the educational level, the characteristics of a coexistence that requires tolerance, empathy and respect for differences, must be considered. Given this educational reality, it seems advisable to explore methodologies that reduce both individualism and competitiveness among students [3]. The implementation of active teaching methodologies, such as project based learning, game-based learning (GBL) or gamification, promotes cooperative learning and may be an appropriate strategy to achieve cooperative learning. They promote the autonomy and involvement of students [4], social interaction, construction of shared knowledge, as well as the development of a culture based on mutual help and support, which fosters a favorable environment for promoting learning for all students [5].

Within these methodologies, this study highlights two (gamification and GBL) that are closely related to each other, as well as to role-playing games (RPGs); however, they have certain differences.

Gamification can be defined as the use of game elements and mechanics in non-recreational contexts [6]. It is frequently supported by online applications, such as Classdojo or Classcraft (Figure 1) [7–9].

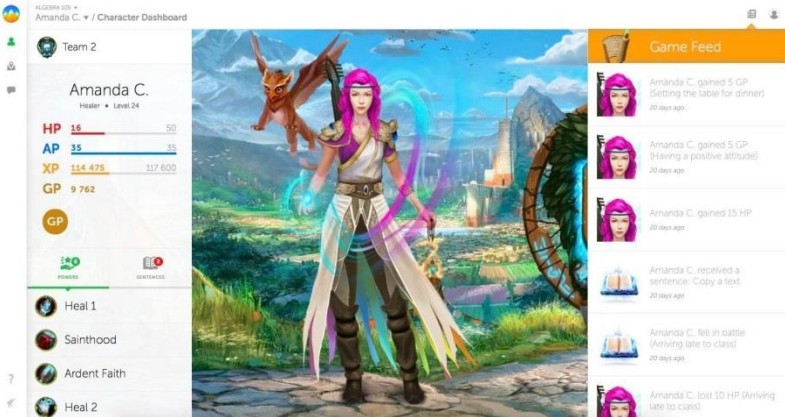

**Figure 1.** Classcraft, a learning management system (LMS) with a gamification taste and strong role-playing game (RPG) relationship (source: classcraft.com).

On the other hand, GBL takes another approach, incorporating games in the educational process [10]. Within GBL, this review focuses on RPGs and their educational use in the Ibero-American context.

RPGs may be defined as a system for creating stories based on rules. They allow a group of players and a game director (also known as Game Master or Narrator) to participate and interact using their imagination to determine what could happen [11]. This shared fiction has its origins in ancient Greek games, although consensus recognizes the 1974 American game "Dungeons & Dragons" as the first modern role-playing game [12,13].

It is important to know that there are different kinds of RPGs that relate to video games and online video games, such as World of Warcraft. The original RPGs, sometimes called tabletop RPGs or pen-and-paper RPGs, were the starting point of their digital relatives, and used mainly speech and imagination; however, technology may be used in some cases for communication, such as "Skype", "Discord", or "Virtual Table Tops" (VTTs) (see Figure 2). RPGs that are played by writing posts in forums or sending emails are called Play by Post (PbP). Some RPGs even require live interaction, similar to theater representations; these are called Live Action Role Play (LARP).

Certainly, RPGs have had bad press [14,15], but judging by the favorable results of investigations, these fears do not have any scientific proof; RPG players do not have suicidal tendencies, have fewer criminal tendencies, and more imagination and empathy [16–19].

Despite this, RPGs have a grim image, much like comics. They are very similar situations, with serious methodological shortcomings (as relates to comics book [20] and video games [21]). The truth is that establishing parallelisms between RPGs and video games can give us the impression that teachers sometimes avoid things that seem to be funny, as if amusement and learning were incompatible. Some authors have insisted on indicating the didactic advantages of video games [22,23], and we cannot forget that many of them are inspired by RPGs. It is also fair to remark that there is research showing the existence of discriminatory or violent values in video games [24].

Once it has been established that this type of game is not harmful, the intention is to analyze how they can be useful in education. Indeed, Ortiz Castells [25] defended RPGs as an active educational methodology and showed several benefits of RPGs in contrast to traditional education. Moreover, Giménez [26] indicated several educational benefits of RPGs:

- They allow access to knowledge in a meaningful way.
- They are considered useful for memorizing tasks.
- They improve mental calculation capacity.
- They promote reading in a playful and recreational way.
- They extend vocabulary.
- They contribute to certain attitudes like development of empathy and tolerance and socialization.

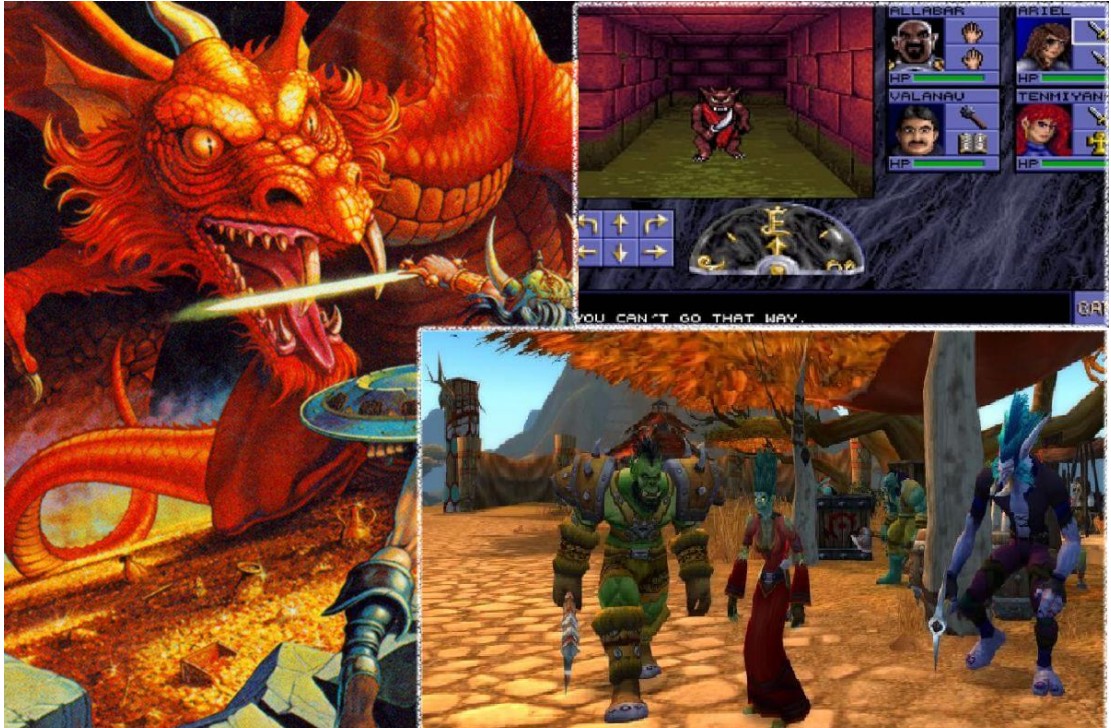

**Figure 2.** First RPG, Dungeons & Dragons (D&D) © WotC. Image from cover of the red box (**left**), a D&D based (Computer Role Playing Game or CRPG), (**top right**) Eye of the Beholder © SSI, image capture from a popular Multi Massive Online Role Playing Game, (**bottom right**) World of Warcraft © Blizzard.

Nowadays, some researchers and teachers around the world have shown deep interest in RPGs as an educational tool [17,18,27–36].

Some possibilities of RPGs in education are:

- Promotion of reading and literature [37].
- History [27].
- English learning [31].
- Physical education [17].

In all these cases (from a GBL perspective), the main strength of RPGs is motivation. RPGs are interactive stories, and students maybe feel better engagement when they feel like the characters of a film or a book. This works for videogames, books, films, RPGs, etc., and learning (experiential learning, or learning by doing).

Additionally, one of the most important advantages of RPGs is their capacity to help us think about morals, ethics and values from a global point of view [38], or more concretely, to think about energy resources, sustainability, and the environment [30,39].

In higher education, several experiences from different subjects/topics may be found below (Table 1).

**Table 1.** Role-playing applications in higher education.

| Authors and Year | Subject/Topic | Sample | Objective |
|---|---|---|---|
| Pérez-López and Rivera-García (2017) [33] | Teacher training | 69 physical activity and sport sciences students | To evaluate teacher training that occurs in a gamification experience, based on students' perceptions of a role-playing game |
| Ortiz-de-Urbina, Medina and de la Calle (2010) [32] | Human resources | About 90 students from three business organization degrees | To analyze the effect and usefulness of role-playing in the field of business organization |
| Diniz and Santos (2019) [29] | Physics | 23 Master's degree Physics students | To present an educational product consisting of the use of the game "In Search of the Nobel Prize" to address topics of atomistics |
| Chiu and Hsieh (2017) [28] | Mathematics | 100 second grade students, with 50 participants each assigned to an experimental group and control group | To explore if significant differences exist in academic performance and learning attitudes between RPG-based assessment and traditional lectures |
| Morales and Villa (2019) [40] | Mathematics | 95 engineering students | To increase the enthusiasm of higher-level students to learn mathematics through adventure school methodology |
| Smutny, Prochazka and Vaculik (2016) [34] | Business and management | 1842 business and economics students | To explore the relationship between managerial skills and managerial effectiveness |
| Wesselow and Stoll-Kleemann (2017) [36] | Geography | 96 students | To explore the diverse potentials of role-playing games (RPGs) in natural resource research and management |
| Soares, Gazzinelli, Souza, and Araújo (2015) [35] | Nursing | 11 undergraduate nursing students | To provide the students with metaphorical experience of problem situations corresponding to the main scenarios of professional activity through a game |
| Ferrero, Bichai and Rusca (2018) [41] | Sustainability | 27 students and 15 researchers | To present a role-playing game designed to foster stakeholder collaboration in water safety plans (WSPs) |
| Campillo-Ferrer, Miralles-Martínez and Sánchez-Ibáñez (2020) [42] | Education and learning | 101 undergraduate education students | To investigate to what extent the popular online gaming platform Kahoot can be used as a creative and effective tool to promote motivation, engagement and meaningful learning |

Most of the research done on role-playing games has been done in higher education, and there is not much research related to professional development in the scientific literature, though there are some exceptions [43–45], mainly using digital tools.

It seems there are a lot of different RPG applications within the context of education. Educational trends such as gamification [41] may change the old negative perspective, incorporating elements of RPGs in applications, e.g., LMS "Classcraft" [7–9], with elements based on classic RPGs, as characters, experience points, levels, etc. The importance of games as a basic part of didactic design is clear, and RPGs can play an important role as an active methodology inside GBL and gamification.

For this, some steps based on other pedagogical activities [18,46] may be useful to apply RPGs inside game-based learning, taking into consideration what to do before, during and after the game.

## 2. Materials and Methods

The main purpose of this study is to analyze the impact of RPGs in Ibero-American education research and to establish the state-of-the-art. Many reviews of scientific articles, within the educational context included in the virtual repository Dialnet, were taken as reference, as well as the bibliographic data offered. The database Dialnet ranks first among European portals and fourth worldwide, with 10,358 journals and 4,822,689 articles [47].

About data search and criteria (Figure 1):

- Data source: Dialnet.
- All searches were confined to the context of education.
- The chain search term used was "role-playing games". Moreover, this chain was used in English, because its translation into Spanish sometimes provokes some confusion between roleplay,

role-playing games and roleplaying. In this case a conservative judgement was followed to avoid terminological confusions.

- The selected articles have been published in journals included in Latindex, and with full text.
- The criteria also included publication dates between 2010 and 2019.
- Those non-related topics have been deleted, because have not any relationship with RPGs.

For analysis and information search purposes, systematic literature review (SLR) [48] and Preferred reporting items for systematic reviews and meta-analyses (PRISMA) [49] (Figure 3) were taken as references, the latter being considered as a perspective closer to mapping. Although SLR was originally oriented to other fields, it is easily adaptable to social sciences [50].

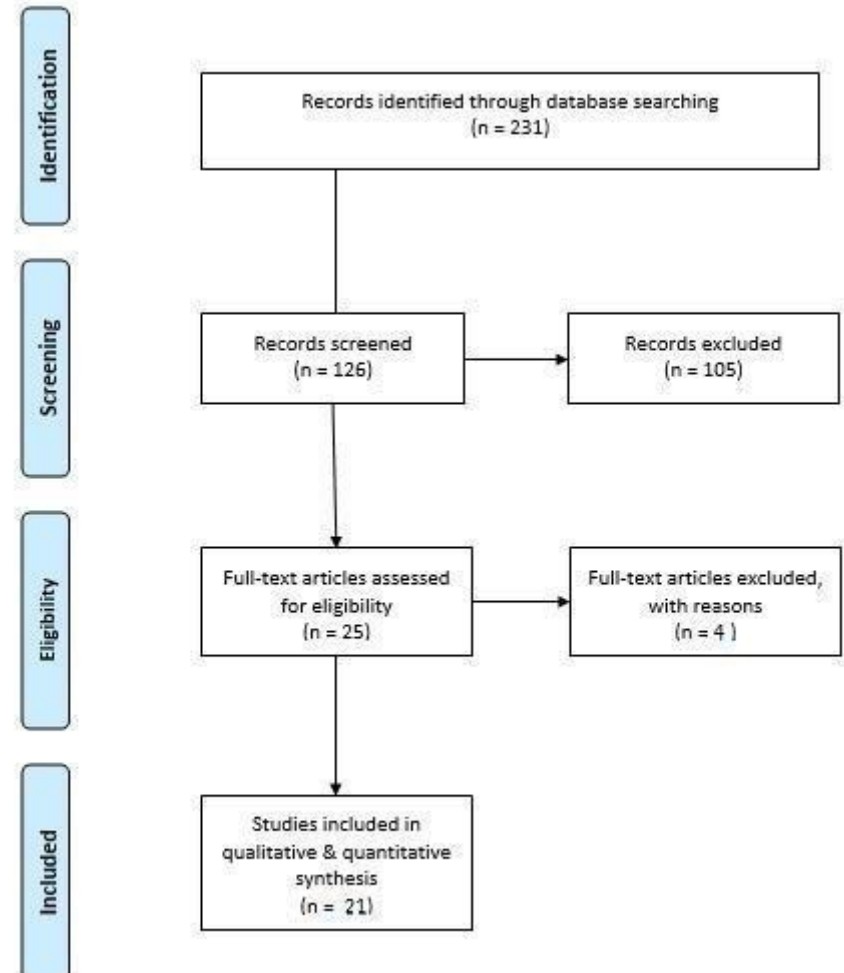

**Figure 3.** Flow diagram indicating phases of review. Source: based on the preferred reporting items for systematic reviews and meta-analyses (PRISMA) model [49].

Our mapping questions are the following:

MQ1. What are the articles published in the last decade in Ibero-America on RPGs?

MQ2. What are the topics of these articles?

MQ3. How many articles are published in Ibero-America about RPGs, by year?

MQ4. How many of the journals in which these articles have been published are indexed in Scopus or Journal Citation Reports (JCR)?

MQ5. Which countries in Ibero-America have published articles about RPGs?

## 3. Results

A total of twenty-five references were found, however, they do not seem to have had a great impact. Four articles were removed once the summary and keywords were revised, as the articles were not related to RPGs. We therefore found a total of twenty-one articles, which are reflected in Table 2, including the year of publication, the authors, and the subject, corresponding to MQ1 and MQ2.

**Table 2.** Articles about RPGs published in Ibero-America (2010—2019).

| Year | Authors | Topic | Google Scholar Citations |
|---|---|---|---|
| 2010 | Ramos-Villagrasa and Sueiro [51] | Player personalities | 3 |
| | Cabrero Sañudo [52] | RPG publications in Spain | 2 |
| | Hernández Carbonell [30] | Attitudes and values | 10 |
| | Roda [53] | Definitions and concepts about RPGs | 16 |
| | Tizón [16] | RPG myths | 3 |
| | Abella-García and Grande-de-Prado [54] | Teacher-in-training perceptions | 10 |
| | Grande-de-Prado and Abella-García [18] | Essay about proposals | 52 |
| | Carbó García and Pérez Miranda [27] | History | 19 |
| | Pérez Miranda and Carbó García [55] | Attitudes and values (gender) | 11 |
| 2013 | Vargas Álvarez [56] | History | 3 |
| | McLean and Griffiths [57] | Video games (review about risks) | 18 |
| 2014 | Rodríguez, Eloy Govea and Serra Hernández [58] | Roleplay (social experiences in early childhood education) | 1 |
| 2016 | Rodríguez-Prieto [59] | Politics and philosophy | 4 |
| | Meza-Jiménez, García-Barrios, Saldívar and Vera [60] | Roleplay (attitudes) | 3 |
| 2018 | Grande-de-Prado [22] | Video games (review about benefits) | 5 |
| | Fernández, Prieto, Alcaraz, Sánchez and Grimaldi [61] | LARP (Physical Education) | 10 |
| | Chacón, Espejo, Martínez, Zurita, Castro and Ruiz [62] | Video games (violence) | 2 |
| | Liesa Orús, Vázquez Toledo, and Latorre Cosculluela [63] | Roleplay (autism) | 1 |
| 2019 | Morales Carbajal and Villa Angulo [40] | Mathematics | 0 |
| | Cruz, Acebal, Cebrián and Blanco [39] | Sustainability | 0 |
| | Mora and Camacho [8] | Classcraft | 3 |

About column "Topic", it is important to remember (from Introduction), that some RPGs had created two kinds of video games: some of the most popular types are computer RPG (CRPG) or online computer RPG (massively multiplayer online role playing game or MMORPG, like "World of Warcraft" or "D&D Neverwinter"); other type of RPGs with live interaction are related to theatre improvisation or psychodrama. These are roleplays, and are very close to Live Action Role Play (LARP), usually longer, with more people, cosplays, a more complex plot, etc., and a great acceptance in North of Europe.

There are some interesting results about the articles found. Trends suggests a more theoretical approach at the beginning of the period studied, becoming more practical at the end, with a key influence from LARPs and Information and Communication Technologies (ICTs). For example, it can be observed that some topics are related to video games, two of them from a negative perspective [57,62], one about educational benefits [22], and finally a recent article about Classcraft [8], very related to MMORPGs [7,9]. Others are related to more physically interactive situations, like roleplaying [58,60,63]

and LARPs [61] with applications for socialization, attitudes, and physical education, and several proposals including history [27,56], attitudes and values [30,55,60], mathematics [40], politics [59], sustainability [39], etc. Additionally, mainly in 2010, several articles had a more miscellaneous nature, with a more theoretical point of view: myths about RPGs [16], editorials and publications [52], profiles of RPG players [51], and definitions and concepts about RPGs [53]. To sum up, there is a great variety of articles and topics.

Our study found 176 Google Scholar citations, determining and interpreting the main measures of center and deviation (mean 8.38, deviation 11.59, and mode 3). Most of the citations belong to a 2010 *Education in the Knowledge Society* (EKS) monograph (128 citations). From a quantitative valuation, 19.05% of references [8,22,53,58] deal with video games and ICTs; 19.05% [58,60,61,63] with LARP/roleplays (differences between them are sometimes difficult to establish); 38.09% [18,27,30,39,40,55,56,59] about practical proposals and experiences in the classroom (without LARPs or ICTs) including history, gender, etc., and the rest deal with diverse issues, often more theoretical, with 23.81% [16,51–54]. This answer corresponds to MQ2.

As shown in Figure 4, most of the articles were published in 2010 (43%). However, all of them (nine in total) belong to a monograph on RPGs from *Education in the Knowledge Society* (EKS), a journal from the University of Salamanca (Salamanca, Spain). Nevertheless, the last two years have shown an increase in the number of articles. The EKS monograph is relevant since in 2010 gamification was not yet considered a trending topic in education, which shows a great interest from the journal, even before games became more popular. This answers MQ3.

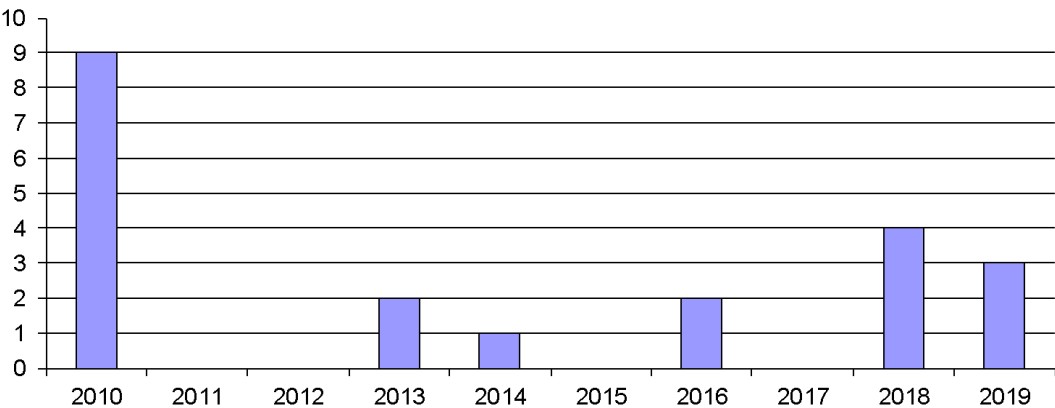

**Figure 4.** Scientific publications in the field of education, about RPGs in Ibero-America, classified by years.

None of the journals where these twenty-one articles have been published is within Journal Citation Reports (JCR), and only one is Q1 or Q2 on Scopus (*Revista Complutense de Educación*); other journals, like *EKS, Educare* or *Aloma,* are within Q3 or Q4 on Scopus, however, not over the entire interval of years. Six journals have never been collected on Scopus. All the publications that are or have been included in Scopus are Spanish journals, except for *Educare*, which is from Costa Rica. *EKS* has more than half of the articles, eleven in total. All the other journals provided only one article each, except for *Revista Complutense de Educación*, with two articles. This answers MQ4.

In terms of geographical distribution, most articles about RPGS come from Spanish journals (sixteen in total), with five of them showing interest in this topic in other countries. Those five articles belong to the following Latin American countries: Colombia, Cuba, Chile, Costa Rica and Mexico. This answers MQ5.

Spanish production stands out for the quantity and quality of publications: fourteen out of sixteen have been or are currently in journals indexed on Scopus. Based on what was found, we can point out the following relevant results on Spanish production:

- Most of the Spanish articles appeared in the *EKS* journal monograph from 2010 [16,18,27,30,51–55]; there are also two other articles published in *EKS* from 2018 [22] and 2019 [8]. Moreover, there is one article published in the journal *Aloma* (2013) [57], and finally, two in the journal *Revista Complutense de Educación*, in 2018 [62,63].

- The analysis shows that the 2010 *EKS* monograph was a relevant milestone, and also indicates a growing interest in the last two years.

- *EKS* journal, with the majority of the production, as well as *Aloma* and *Revista Complutense de Educación* journals are indexed in Scopus, although only the last one belongs to Q1 or Q2 on Scopus. None of the journals have been indexed in JCR and two other journals (with one article each) have never been indexed in JCR or Scopus.

- Regarding the main topic of the article, four references deal with video games/ICTs, two with LARP/RPGs and the rest (the majority of them) deal with RPGs, although with different and varied nuances.

The previous results answer questions MQ2, MQ3, MQ4 and MQ5, analyzing exclusively the Spanish situation.

## 4. Discussion

The aim of the present study is to represent the current situation related to RPGs in Ibero-American educational research. The systematic literature review (SLR) developed in this research and the results obtained allow us to state that there is a certain interest in these games within education; however, they are not a subject of wider significance in the Ibero-American educational scientific literature.

Twenty-one articles were found throughout Ibero-America, of which sixteen belong to Spanish journals; almost all of the journals have a certain impact. Inside the Spanish production, the *Education in the Knowledge Society* (EKS) journal has a great impact, with eleven articles, followed by the journal *Revista Complutense de Educación*, with two articles. Without the monograph from *EKS* in 2010, interest in the literature would not have been so strong. RPGs would have been found, especially compared with popularity of gamification [41,42,64], for example, with the use of Classcraft [7–9], or GBL. Both tendencies have a strong relationship with RPGs, a methodology with a huge potential in education [59].

Perhaps, venturing some hypotheses, the outcomes derive from the negative publicity derived from the alarmist vision that arose in the 1980s in the United States about the danger of RPGs [16] and practical problems regarding their educational application [18]. Despite their bad reputation, RPGs have had a key influence in some educational methodologies. An approach from CRPGs or MMORPGs seems more accessible or attractive to 21st-century students, as the trending topic suggests, who are more familiar with these variants of classic RPGs, due to the growing gamification integrated into educational practices and tasks, and its wide-ranging advantages and potential, derived from its diverse, motivational and playful format. In this way, the acquisition of knowledge is favored, but especially students' motivation to learn.

This review can serve as a basis to go deeper into the RPG topic within the Ibero-American environment. Future work may include extending subsequent research through analysis of scientific contributions in other languages or other non-educational areas, allowing further comparisons. Moreover, it is necessary to emphasize the need for critical analysis of RPGs, so that in addition to the characteristics, the difficulties and limitations found in this type of training action can be known. The previous study may be useful to establish future lines of action, allowing the optimization of the entire chain between the design and implementation of RPGs in classrooms.

As mentioned previously, RPGs and gamification have several strong bonds [7–9,41,42,64]. This is another important issue regarding RPGs and education: how roleplaying games may be used in educational settings, inside game-based learning? The following steps, based on other pedagogical activities [18,46], are independent of what kind of RPG variation is used: tabletop RPG/pen and paper RPG; RPGs using ICT tools like Discord, Skype or VTT; Play by Post (PbP) RPGs; or Live Action Role Play (LARP).

(a)　Before bringing it up to the classroom. The first thing to do is to get to know the game and its settings. A record in our database should also be prepared, which will be filled up at the end, since we have to add both progress and real circumstances once the game has finished. It is also advisable to make other preparations, such as creating a conceptual map of the most important contents and concepts, planning the previous and subsequent activities, thinking about the aim, estimating the time required for the game, having a content scheme, preparing materials, creating a favorable atmosphere, understanding a simple and efficient game system (there are a lot of free games nowadays, as well as generic systems developed at the commercial level), and understanding some of the narrative resources used in RPGs (these are available in many web pages and almost in every RPG manual).

(b)　In class, but before playing. Several strategies might be chosen according to pupils' age, skills, or knowledge. For example, collecting information from their previous knowledge; doing activities prior to beginning game play, e.g., reading or watching videos; introducing pupils to general game rules; forming game groups, including observers if necessary.

(c)　During the game. Students may apply previous analyses to comment or detect relevant aspects or mistakes, which can be analyzed in depth.

(d)　After the game. The teacher may look for analogies, going back to quandaries or covered topics, and asking the pupils about their sensations during the game, e.g., what attracted their attention, and may propose later brief investigations or treasure hunts to settle doubts, among other possibilities.

These ideas summarize an appropriate structure for using RPGs in the classroom. Beyond these recommendations, there are more questions about RPGs & education. Most of the articles are about educational games (purpose-built), in a similar way to serious games, but there are many RPG games that can be used for educational purposes, for example in literature [37] or history [18]. Gamification can easily be introduced to the classroom, but the use of games (commercial or self-made) has a more complex nature [18].

To wrap up, rather than a definitive conclusion on future directions for RPGs and education, we hope that this study serves as an opening for exploring the possibilities of RPGs inside classrooms. In this sense, teachers can become aware of the possibilities of adapting RPGs in the teaching and learning process, paying attention to the number of students, ages, content, objectives and/or competences to be developed.

**Author Contributions:** Conceptualization: M.G.-d.-P., R.B., S.G.-M. and V.A.-G.; methodology: S.G.-M. and V.A.-G.; collected the data: M.G.-d.-P. and R.B.; writing—original draft preparation: M.G.-d.-P.; writing—review and editing: M.G.-d.-P. and R.B.; supervision: S.G.-M. and V.A.-G. A previous conference paper called "RPGs as an educational tool in Ibero-America" (CUICIID 2019 Conference) has been taken as a basis, fulfilling the requirements of Sustainability (see cover letter). All authors have read and agreed to the published version of the manuscript.

**Funding:** This research received no external funding.

**Conflicts of Interest:** The authors declare no conflicts of interest.

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
