# Peer review of "Mapping Role-Playing Games in Ibero-America: An Educational Review"

_sustainability, doi:10.3390/su12166298_

Round 1

Reviewer 1 Report

Overall this is a well-structured systematic review that describes the process clearly. While there is a fairly small final set of studies, it is timely and capturing the field at a critical time in terms of early growth.

As the focus is on educational application of role-playing games, there was too much detail given on the historical negative views of role-playing games in general. This section could be made more concise and the connections with school contexts made more explicit.

Table 1 is very interesting in terms of the Higher Education applications. I would like to see more detail on the use of roleplaying in professional development contexts.

With the systematic review terms, it is unclear exactly which search-terms were used. Was it just "role-playing games"? Or were synonyms such as RPGs, roleplaying, role playing games, etc used. What efforts were made to capture games or teaching strategies that are not labelled as roleplaying games but feature an element of characterisation and gamification?

In the discussion, it would benefit from more detail on how roleplaying games are used in educational settings. Can a distinction be made between educational games (purpose-built) and games being used for educational purposes? This could connect with a broader discussion of the use of gamification in education settings - is it being used to engage pupils and enable 'stealth teaching' and implicit learning (as is often the case in language teaching games) or is roleplaying being used as simulation?

Reviewer 2 Report

The paper presents a Systematic Literature Review (SLR) of published papers on methodologies that include role-playing games (RPGs) features in Education. The analysis is restricted to the decade 2010-2019 for Spain and Latin America. The review is performed through searching and filtering from the bibliographic database Dialnet and is structured following the PRISMA guidelines. The interest in this contribution lays in the aim of shedding some light on a topic which has not been explored extensively in the educational literature. What is found is that only 21 papers fit the selection filter utilised, out of which 9 belong to the same monograph.

The paper can be of interest to a general audience because it focuses on a topic potentially useful for education which has not been investigated much, applyin a standard PRISMA approach also in the use of the flow diagram. However, the results of the SLR are not easily interpretable since the large majority of papers belong, as also pointed out by the authors, to a single monograph publication and therefore the conclusions about the dynamics of the interest in the literature cannot be so strong.

In what follows, I list some specific comments which I suggest to take into account for a potential review:

  1. It could be advisable not to overstate the purpose of the present study by rephrasing line 6 in the abstract which refers to the "impact" of RPGs. The same correction is suggested for line 1 in section "Discussion". Instead, one can use a different wording such as "report the state of the art..." or "represent the current situation...";

  2. In section "Materials and Methods", in bullet point 2, it is stated that the search is confined to the context of Education. However, it could be good to consider specifying how the selection has been performed since it does not seem to be an option to filter for research fields on the Dialnet filter tool;

  3. In the same section, bullet point 3, it is specified that the chain search terms used is ’role-playing games’. It can be good to know if also synonyms have been considered, such as RPGs, role playing games, juego de rol, JDR, and if not, and the search results are changing, it can be the case to add them in the analysis;

  4. In Table 2, column "Topic", there are some entries which are categorized under a very general and broad label, such as "Video game" or "Roleplay". Having more detailed descriptions of the topics could be advisable for a better understating;

  5. The quality measure proposed (Scopus or JCR indexes) cannot be used for the majority of cases, therefore one may want to consider alternatives such as number of citations in Google Scholar (and also add this information in Table 2);

  6. The final paragraph in page 4 seems to go out of the scope of the paper. The idea of suggesting a structure for using RPGs in classrooms is certainly interesting but does not link with the objective defined by the authors. Therefore, one may want to opt for removing this paragraph and the connected citations or either adding in the "Discussion" section a briefer reference to the issue without trying to solve it in this framework;

  7. Lastly, I comment on the closeness of the salient topic treated (learning and education) to the fields covered in the MDPI "Education Sciences" journal (whose scope involves "learning and teaching", "special education", "educational technology systems ", etc.), that could fit this study’s publication more than the MDPI "Sustainability" journal (whose most close subject area is only "Education and awareness of sustainability").

  8.  

Reviewer 3 Report

This article is a nice presentation of the main research around RPG in IberoAmerica. It identifies the key points achieved in the field and open new research lines.

Round 2

Reviewer 1 Report

Very good and thoughtful amendments which I believe enhance the paper. The discussion has been both widened and deepened effectively. An excellent review that I believe will support future research in a developing topic. 

Reviewer 2 Report

I am fine with the revision.